

# Reliability and validity of an app-assisted tissue compliance meter in measuring tissue stiffness on a phantom model

Andreas Brandl[1,2,3], Eda Acikalin[2,4], Katja Bartsch[2,5], Jan Wilke[6] and Robert Schleip[2,3]

[1] Department of Sports Medicine, Institute for Human Movement Science, Faculty for Psychology and Human Movement Science, University of Hamburg, Hamburg, Germany
[2] Department of Sport and Health Sciences, Conservative and Rehabilitative Orthopedics, Technical University of Munich, Munich, Germany
[3] Department for Medical Professions, Diploma Hochschule, Bad Sooden-Allendorf, Germany
[4] Faculty of Physical Therapy and Rehabilitation, Hacettepe University, Ankara, Turkey
[5] Department of Sport Science and Sport, Friedrich-Alexander Universität Erlangen-Nürnberg, Erlangen, Germany
[6] Department of Movement Sciences, University of Klagenfurt, Klagenfurt, Austria

Corresponding author
Robert Schleip,
robert.schleip@tum.de

## ABSTRACT

**Background:** Most methods for soft tissue stiffness assessment require high financial resources, significant technical effort, or extensive therapist training. The PACT Sense device was developed to be used in a wide range of applications and user groups. However, to date, there are no data on its validity and reliability. The aim of this study was to investigate the validity and reliability of the PACT device.
**Methods:** A polyurethane phantom tissue model (PTM) mimicking the mechanical properties of the fascia profunda and the erector spinae muscle was used. Stiffness measurements with PACT were conducted by two independent investigators. For construct validity, correlations were calculated between the known stiffness of the PTM and values obtained with PACT. For concurrent validity, we determined the association between the PACT values and additional measurements with the established MyotonPRO device. To estimate interrater and intrarater (two measurements with an interval of 7 days) reliability, we used the intraclass correlation coefficient (ICC).
**Results:** Correlation analysis (PTM/PACT) revealed very high concurrent validity ($r = 0.99$; $p < 0.001$), construct validity (PACT/MyotonPRO) was 0.87, $p < 0.001$. Both, interrater reliability (ICC = 0.85; $p = 0.036$) and intrarater reliability were good (ICC = 0.89; $p < 0.001$).
**Conclusions:** The PACT provides valid and reliable stiffness measurements in tissue phantoms. Further studies in humans are needed to confirm its physiometric properties under *in vivo* conditions.

# INTRODUCTION

Soft tissue stiffness is a mechanical property defined as the resistance of biological materials to an external deforming force. Changes in myofascial tissue stiffness, which in

this article is understood as the combination of a muscle, its soft tissue components (endo-, peri- and epimysium), and the overlying fascial structures such as the thoracolumbar fascia, are associated with acute and chronic pain, micro- and macro-injuries, musculoskeletal disorders, and carcinogenesis (*Langevin et al., 2011*, *2016*; *Schleip et al., 2012*; *Kuo et al., 2013*; *Brandl et al., 2022*). Furthermore, it could play an important role in the development and maintenance of sports performance (*Arampatzis et al., 2001*; *Kalkhoven & Watsford, 2018*; *Moran et al., 2023*). The applications of stiffness measurement are therefore manifold, ranging from therapy monitoring or quantification of training effects to stiffness-dependent modifications of manual tissue intervention techniques (*Ajimsha, Al-Mudahka & Al-Madzhar, 2015*; *Kalkhoven & Watsford, 2018*; *Zügel et al., 2018*; *Moran et al., 2023*).

Traditionally, a therapist palpates and quantifies tissue stiffness based on several years of professional training. However, the reliability of palpation-based tissue assessments has been discussed inconsistently in previous works (*Nolet et al., 2021*; *Brandl, Egner & Schleip, 2022*). Therefore, in addition to high-priced imaging technologies such as shear wave ultrasound or magnetic resonance elastography, lower-cost mechanical devices have been developed to evaluate the compressive stiffness of tissues (*e.g.*, MyotonPRO, IndentoPro, Shore durometer). However, these devices are not widely used in manual therapy practice and more commonly seen in clinical settings because they require a trained operator or/ and the devices are still relatively costly.

The PACT device, a hand-held stiffness measurement tool, combines a user-friendly app-based device operation and an indentometric probe unit. According to the developer, it can be used by trainers, coaches, and therapists for self-monitoring of muscular stress states in sports as well as for myofascial diagnostics. However, unlike the MyotonPro, which showed good correlations with ultrasound shear wave elastography (*Kelly et al., 2018*), there is no validity or reliability data for the PACT.

In a previous study, a polyurethane phantom tissue model (PTM) mimicking the tissues of the lumbar region with known viscoelastic properties was used to examine reliability and validity of different stiffness measurement tools (*Bartsch et al., 2023*). This approach has been recommended by several authors in previous studies (*Oflaz & Baran, 2014*; *Sohirad et al., 2017*; *Wilke et al., 2018*).

The aim of this work was to evaluate the criterion validity of the PACT on one phantom layer (mimicking the erector spinae muscle) and two combined phantom layers (mimicking the erector spinae muscle and the overlying fascia profunda). This setup was used because perpendicular indentometric stiffness measurements are not able to exclusively measure soft or muscle tissue structures, and biomaterial *in vivo* is always a combination of both. In addition, to determine concurrent validity, measured values were compared with an established device (MyotonPRO). Furthermore, the interrater and intrarater reliability were analyzed. Our hypotheses were therefore that the PACT would achieve high criterion and concurrent validity as well as good intra- and interrater reliability.

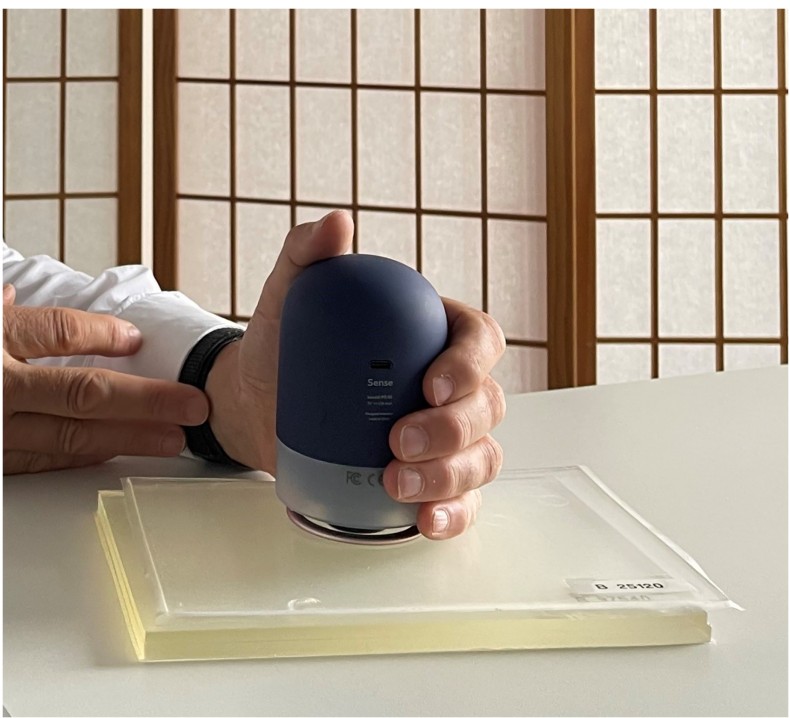

**Figure 1** PACT Sense on the two-layerd phantom tissue model.

## MATERIALS AND METHODS

This was a validity and reliability study with two blinded investigators conducting stiffness measurements using a PTM. It followed the Guidelines for Reporting Reliability and Agreement Studies (GRRAS; *Kottner et al., 2011*).

### Two-layered phantom tissue model

A polyurethane gel pad PTM (21 cm × 31 cm; Technogel Germany GmbH, Berlingrode, UK) was used, which was developed in advance and implemented in a previous study mimicking the tissues of the human lumbar region (*Bartsch et al., 2023*). Briefly, the two-layer PTM consisted of the fascia profunda and the erector spinae muscle. Through a literature review, the typical thickness and stiffness of each of these layers were determined and a thickness of 3 and 10 mm, respectively were chosen (Fig. 1; *Langevin et al., 2011*; *Nair et al., 2016*; *Moreau et al., 2016*; *Lohr et al., 2018*). Because stiffness varies in living organisms, each layer was manufactured with different stiffness parameters, specified in Shore OOO and converted to Young's modulus in kPa (*Mix & Giacomin, 2011*). Table 1 shows the varying stiffness parameters of the layers.

### PACT sense

The PACT Sense (Impact Biosystems Inc., Boston, MA, USA) is a digital stiffness measurement device consisting of an instrument housing and an indentation probe

**Table 1 Polyurethan layer stiffness variation.**

| Hardness (Shore OOO) | Young's modulus kPa |
|---|---|
| 15.00 | 24.43 |
| 20.00 | 30.34 |
| 25.00 | 37.36 |
| 30.00 | 45.81 |
| 35.00 | 56.08 |
| 40.00 | 68.75 |
| 45.00 | 84.62 |
| 50.00 | 104.87 |
| 55.00 | 131.30 |
| 60.00 | 166.80 |

**Table 2 MyotonPRO validity and reliability from previous study.**

| | Validity | | | Interrater reliability | | |
|---|---|---|---|---|---|---|
| PTM layer | Cor | Linear regression formula | $R^2$ | $ICC_{(2,2)}$ | 95% CI | MDC% |
| 3 mm | 0.94* | (0.0061 + 0.0757 * RV) | 0.88* | 0.98* | [0.86–0.99] | 2 |
| 10 mm | 0.91* | (0.0076 + 0.0181 * RV) | 0.84* | 0.94* | [0.61–0.99] | 1 |

Note:
Data were collected from a multilayered phantom tissue model in a study by *Bartsch et al. (2023)*. Cor, Pearson product-moment correlation; RV, real values, stiffness of the tissue phantom; MDC, minimal detectable changes. Significant at the level *<0.001.

(Fig. 1). *Via* mechanical pulses into the skin surface with the probe, while keeping the contact pressure of the housing constant, the device collects data on the response of the underlying tissue at three different indentation depths. The collected data is then further processed forward with algorithms to calculate the stiffness (N/m), which is displayed to the user through a mobile application.

## MyotonPRO

For cross-validation, the MyotonPRO (MyotonAS; Tallinn, Estonia) was used because its validity for assessing viscoelastic properties of myofascial tissue, especially stiffness, has been demonstrated in recent studies (*Feng et al., 2018*; *Bartsch et al., 2023*). In addition, the measurement principles appeared to be very similar to indentation by a probe with a defined series of mechanical impulses. The validity and reliability measured on the multilayer PTM according to *Bartsch et al. (2023)* are listed in Table 2.

## Measurements

The one-layer setup consisted of a 10-mm-thick PTM mimicking the erector spinae muscle. Tissue stiffness changes were simulated using 10 different stiffness configurations of the gel pad (Table 1) and measured with the devices.

For evaluation of a two-layer setup, the material phantoms were placed on top of each other according to their natural order (1st layer, fascia profunda; 2nd layer, erector spinae muscle). To mimic stiffness changes in the PTM, four different layer variants representing four different stiffness parameters (24.43, 45.81, 84.62, 166.8 kPa) were exchanged one after the other, with the 1st layer remaining in a measurement set configuration (example: the 4 gel pad variants for the 2nd layer were exchanged and measured individually with the device, while the 1st layer remained in a stiffness configuration; analogous procedure was followed for the three other stiffness variants of the 1st layer). Therefore, a total of 32 measurements were carried out in different stiffness configurations with the two-layer setup. The measurements were performed by two examiners in a blinded manner, *i.e.*, the examiners did not know the stiffness parameters of the individual gel pads. The device was placed perpendicular to the gel pad approximately in the center of the gel pad by eye. For each gel pad configuration, three consecutive measurements were obtained by the respective rater and averaged. For intrarater reliability, Rater 1 performed a first series of measurements and a second one week later. Both raters had been familiar with the MyotonPro for several years, but were novices in using the PACT Sense. Therefore, they had no prior training and only followed the app-guided instructions in the user manual.

## Statistical analysis

All descriptive data are means ± standard deviation (SD). Criterion validity, and interrater reliability (for both SAT) as well as intrarater reliability (for the PACT Sense) were assessed on the one-layer setup (PTM mimicking the erector spinae muscle). Further, a second criterion validity measurement was performed on the two-layer setup (PTM mimicking the fascia profunda and the erector spinae muscle). Changes in device-measured stiffness and the corresponding Young's modulus of the PTM were assessed using Pearson product-moment correlation for data that met the criteria for parametric testing or, if not, Spearman rank correlation. Subsequently, the device measurement was used as a predictor for the Young's modulus of the PTM in a linear regression analysis (with log 10 transformation for non-parametric variables). The resulting coefficients were interpreted as 'low' (0.3 to 0.5), 'moderate' (0.5 to 0.7), 'high' (0.7 to 0.9) or 'very high' (0.9 to 1.0; *Mukaka, 2012*). Intraclass correlation coefficient (ICC) estimates between PTM and device measurement and their 95% CI were calculated based on a two-way mixed-effects model with absolute agreement. Not normally distributed data, were log-10 transformed. Resulting ICC values were interpreted according to *Koo & Li (2016)* as 'poor' (<0.50), 'moderate' (0.50 to 0.75), 'good' (0.75 to 0.90) and 'excellent' (>0.90). For relative reliability (*Furlan & Sterr, 2018*), the corresponding standard errors of measurement (SEM) were estimated using the formula (*Schmitt & Di Fabio, 2004*).

$$SEM = SD * \sqrt{(1 - ICC)}$$

In order to calculate the SEM independently of the different units of the devices, the percentage SEM (SEM%) was defined as follows:

**Table 3  Validity measurements on a phantom tissue model.**

| Device | 1st Layer (kPa) | Cor | p-value | Linear regression formula | $R^2$ | p-value |
|---|---|---|---|---|---|---|
| PACT sense | –* | 0.99 | <0.001 | (0.235–365.45 * RV) | 0.98 | <0.001 |
| | 24.43 | 0.59 | 0.063 | (0.194–274.04 * RV) | 0.34 | 0.126 |
| | 45.81 | 0.71 | 0.025 | (0.189–291.11 * RV) | 0.50 | 0.050 |
| | 84.62 | 0.93[1] | <0.001 | (0.096–125.93 * RV) | 0.69 | 0.011 |
| | 166.8 | 0.79 | 0.009 | (0.230–395.76 * RV) | 0.63 | 0.019 |
| MyotonPRO | –* | 0.87 | <0.001 | (0.525–407.06 * RV) | 0.75 | 0.001 |
| | 24.43 | 0.95[1] | <0.001 | (13.5–461.70 * RV) | 0.90 | <0.001 |
| | 45.81 | 0.87 | 0.002 | (8.65–347.70 * RV) | 0.76 | 0.005 |
| | 84.62 | 0.95 | <0.001 | (18.2–635.50 * RV) | 0.90 | <0.001 |
| | 166.8 | 0.89 | 0.002 | (9.14–380.47 * RV) | 0.79 | 0.003 |

**Notes:**
Cor, Pearson product-moment correlation.
[1] Not normal distributed data were instead calculated with the Spearman's rank correlation.
* One-layer setup (validity data were calculated based on the 2nd layer with 10 mm thickness and varying stiffness; the other measurements show a two-layer setup combining the 1st layer with a 2nd layer with varying stiffness; stiffness values for the second layer were 24.43, 45.81, 84.62, 166.8 kPa, respectively, corresponding to the first layer). RV, real values, stiffness of the tissue phantom; MDC, minimal detectable changes.

$$SEM\% = \frac{SEM}{\bar{x}} * 100$$

Here $\bar{x}$ is the mean for all observations. The minimal detectable change (MDC) was estimated by reference to the SEM using the formula (*Furlan & Sterr, 2018*):

$$MDC = 1.96 * \sqrt{2} * SEM = 1.96 * \sqrt{2} * SD * \sqrt{(1 - ICC)}$$

The percentage MDC (MDC%) was defined as (where $\bar{x}$ is the mean of all observations):

$$MDC\% = \frac{MDC}{\bar{x}} * 100$$

Bland–Altman plot with limits of agreement was created to provide additional visual information about limits of agreement between raters (*Bland & Altman, 1995*).

All analyses were performed using Jamovi 2.3 (The jamovi project, https://www.jamovi.org).

# RESULTS

Correlation and linear regression analysis revealed very high concurrent validity between the Young's modulus of the PTM in a one-layer setup and the PACT device (r = 0.99, $p < 0.001$; $R^2 = 0.98$, F(8) = 309, $p < 0.001$; Table 3; Fig. 2). For stiffness changes in a two-layer setup of the PTM there were also significant correlations except for the softest PTM layer (24.43 kPa), with effect sizes ranging from to 0.71 to 0.93, all $p < 0.025$ (Fig. 3; Table 3).

Construct validity between the PACT and the MyotonPRO in a one-layer setup showed a correlation of r = 0.87, $p < 0.001$ and the linear regression was $R^2 = 0.75$, F(8) = 20.6,

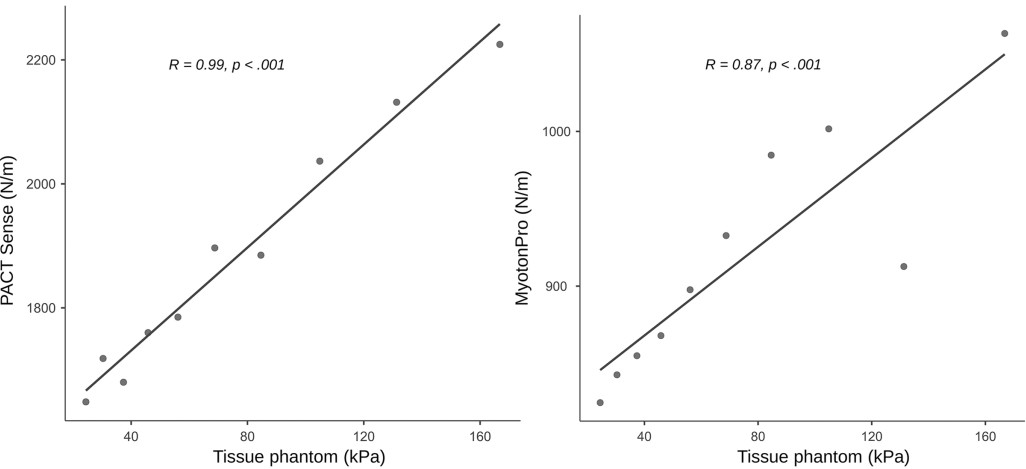

**Figure 2 Concurrent validity measurements on a phantom tissue model.** The figure shows the scatter plots with regression lines and the Pearson product-moment correlation based on a single-layer setup with 10 mm thickness.

$p = 0.002$ (Table 3; Fig. 2). In the two-layer setup of the PTM there were also significant correlations with effect sizes ranging from 0.87 to 0.95, all $p < 0.002$ (Table 3).

The interrater reliability for the PACT was good (ICC$_{(2,2)}$ = 0.85; $p = 0.36$) and for the MyotonPRO excellent (ICC$_{(2,2)}$ = 0.99; $p < 0.001$; Table 4). Bland-Altman plots for interrater reliability showed a moderate leftward shift for the PACT, indicating somewhat larger interrater differences when measuring lower stiffness values. The plot for the MyotonPRO showed that the points were scattered in an unbiased pattern. All the measurement points of the device were within the limits of agreement (Fig. 4).

The intrarater reliability for the PACT was good (ICC$_{(2,2)}$ = 0.89; $p < 0.001$; Table 4).

## DISCUSSION

To our knowledge, this is the first technical validity and reliability study of the PACT device and the results suggest that it could be a valid and reliable tool for measuring stiffness in myofascial tissue layers mimicked by a PTM. In a single-layer setup, the instrument showed very high correlations between artificial PTM stiffness changes and the PACT values. The regression analyses of the PACT measurements explained 97.5% of the stiffness changes. With a two-layer setup, the devices showed moderate to very high correlations, except for the PACT, where the softest layer was on top but only showed a trend ($p = 0.063$). This slight decrease in accuracy for softer PTM strata was also seen in the Bland-Altman analysis of interrater reliability. This could limit the ability to monitor tissues in conditions associated with softening (*e.g.*, swelling and edema). Intrarater and interrater reliability were good overall for the PACT Sense and interrater reliability was excellent for the MyotonPRO, indicating that both devices are capable of detecting changes in tissue stiffness, with slightly higher accuracy for the MyotonPRO device. The results are in line with previous studies on the validity and reliability of the MyotonPRO, which also showed very high correlations with the real stiffness values (*Bartsch et al., 2023*) and

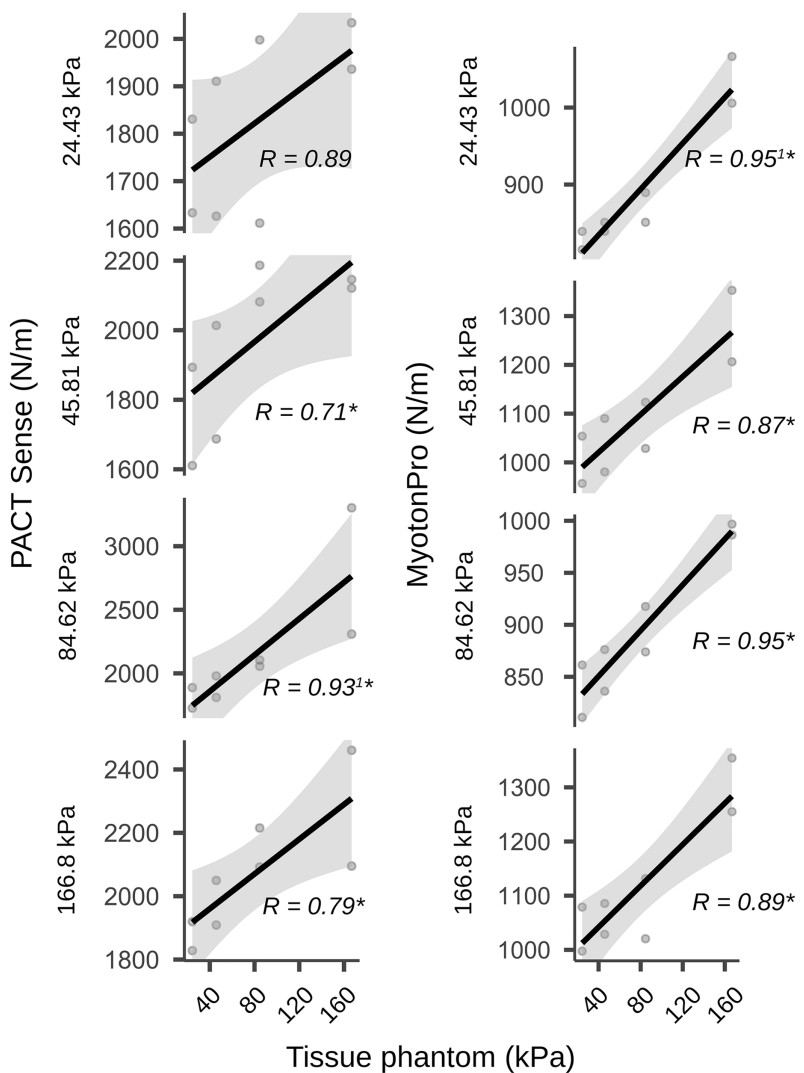

**Figure 3 Validity measurements on a two-layer phantom tissue model.** The figure shows the scatter plots with regression lines and the Pearson product-moment correlation coefficients. The plots show the two-layer setup in which the first layer was combined with a second layer of different stiffness (the stiffness values for the second layer were 24.43, 45.81, 84.62 and 166.8 kPa). Therefore, a total of 32 measurements were carried out in different stiffness configurations. Gray shadows show the 95% confidence intervals. [1]Non-normally distributed data were calculated using the Spearman rank correlation instead of the Pearson product-moment correlation; An asterisk (*) indicates significant at the $p < 0.05$ level; AU, arbitrary units.

**Table 4 Reliability measurements.**

| Device | Typ | ICC | 95% CI | P-value | SEM | SEM% | MDC | MDC% |
|---|---|---|---|---|---|---|---|---|
| PACT sense | Inter | 0.85 | (−0.14 to 0.97) | 0.036 | 93.65 N/m | 5.19 | 260 N/m | 14.4 |
| | Intra | 0.89 | (0.62–0.97) | <0.001 | 82.46 N/m | 4.73 | 224 N/m | 12.8 |
| MyotonPRO | Inter | 0.99 | (0.90–0.99) | <0.001 | 7.16 N/m | 0.78 | 19.85 N/m | 2.1 |

**Note:**
The reliability measurements are based on a single-layer setup with stiffness values of 24.43, 45.81, 84.62 and 166.8 kPa. MDC, minimal detectable changes; ICC, inter- and intrarater reliability; CI, confidence interval; SEM, standard error of the mean; %, dimensionless percent values.

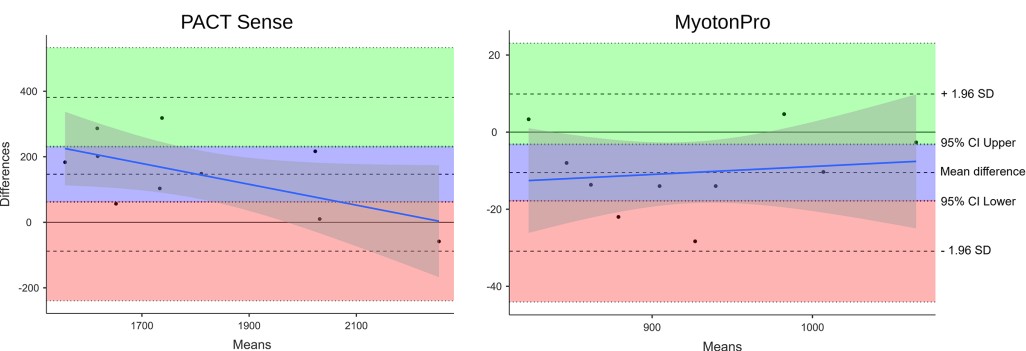

**Figure 4 Bland-Altman plot of the interrater reliability.** The reliability measurements are based on a single-layer setup with stiffness values of 24.43, 45.81, 84.62 and 166.8 kPa. The PACT plot shows a moderate left shift, indicating somewhat larger interrater differences when measuring lower stiffness values. The MyotonPRO diagram shows that the points are scattered in a rather unbiased pattern. All measurement points are within the limits of agreement. The blue line shows the proportional bias. Gray shading shows the 95% CI. SD, standard deviation; CI, confidence interval.

excellent intra- and interrater reliability (*Kelly et al., 2018*; *Lohr et al., 2018*; *Bartsch et al., 2023*). The manufacturer of the MyotonPro only provides prices for the device on request. However, the average cost of the three devices used in the departments participating in the study is $5,000 USD. At $599 USD, the PACT Sense costs only a tenth of that, making it affordable for the daily practice of trainers, coaches, or therapists (*Impact Biosystems Inc, 2024*). Soft tissue assessments with the PACT Sense or the MyotonPRO may have a wide range of applications. They could be useful in monitoring the effects of therapy or training (*e.g.*, to control the recovery process), the diagnosis of musculoskeletal or neurological diseases, or in the stiffness-dependent modification of manual tissue intervention techniques. In the latter case, this could be particularly meaningful for manual therapists, as it is difficult to establish reliability and consistency between practitioners in manual palpation. Results vary widely, and reliability in detecting changes in tissue stiffness is controversial (*Ajimsha, Al-Mudahka & Al-Madzhar, 2015*; *Zügel et al., 2018*). Given the possible role of myofascial tissue in maintaining health and human posture, an affordable, valid, and reliable device for detecting changes in stiffness could be particularly useful (*Nolet et al., 2021*; *Brandl, Egner & Schleip, 2022*).

Some limitations need to be discussed. First, due to production limitations, the PTM mimicking the myofascial tissue could not be a 100-fold prosthetic mechanical double, as the thickness of the fascia profunda and erector spinae had to be adjusted to meet the technical requirements. We are convinced that this shortcoming is compensated by the stability of the mechanical properties, which allowed for an experimental validity and reliability check of the devices in this study. The PTM proved to be suitable for measurements of compressive stiffness, the latter being characterized by a force acting perpendicularly on the material. However, the design of the model did not allow investigations of shear strains, which would involve a force applied laterally to the medium (*Langevin et al., 2011*; *Krause et al., 2016*; *Brandl et al., 2023a*, *2023b*). In both diagnostic measurements and therapeutic applications, shear mobility between tissue layers can play a

significant role in determining muscle and fascia health. Another issue relates to probe placement. Due to its flat surface, the PTM allowed an easy positioning of the devices. Yet, the contour of the human body is markedly different, including bony prominences, smaller contact areas and different shapes. Accordingly, we expect lower validity and reliability values (*e.g.*, like an between day ICC < 0.90 found for the MyotonPro device (*Lohr et al., 2018*)) and further *in vivo* studies involving symptomatic and healthy subjects are strongly recommended.

## CONCLUSIONS

The PACT device represents an easy-to-handle instrument with high validity and good reliability for assessments of soft tissue. However, its ability to detect very low stiffness may be limited. While both devices could confidently be used in clinical settings, the MyotonPRO may be preferrable if maximal precision and reproducibility are required.

### Funding

The authors received no funding for this work.

### Competing Interests

Robert Schleip is an independent non-permanently employed scientific advisor to Impact Biosystems Inc.

### Author Contributions

- Andreas Brandl analyzed the data, prepared figures and/or tables, authored or reviewed drafts of the article, and approved the final draft.
- Eda Acikalin conceived and designed the experiments, performed the experiments, prepared figures and/or tables, authored or reviewed drafts of the article, and approved the final draft.
- Katja Bartsch conceived and designed the experiments, authored or reviewed drafts of the article, and approved the final draft.
- Jan Wilke analyzed the data, authored or reviewed drafts of the article, and approved the final draft.
- Robert Schleip conceived and designed the experiments, performed the experiments, prepared figures and/or tables, authored or reviewed drafts of the article, and approved the final draft.

### Data Availability

The raw measurements are available in the Supplemental File.

### Supplemental Information

Supplemental information for this article can be found online at http://dx.doi.org/10.7717/peerj.17122#supplemental-information.

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
