# Peer review of "Reliability and validity of an app-assisted tissue compliance meter in measuring tissue stiffness on a phantom model"

_PeerJ, doi:10.7717/peerj.17122_

## Round 0.1 · original submission · Major Revisions

The authors should provide stronger justification of using the PACT compared to the MyotonPro. Overall, I commend the authors for their study design, as it a strong comparative investigation of two technologies (one new to the market) using phantoms. Most of my concerns below are clarifying points, and not overall weakness in their approach.

Reviewer 1 ·

Basic reporting

The submitted paper aims to validate a novel method, the PACT device, to assess soft tissue mechanics. The authors propose the use of the PACT device may provide more accessible clinical measurements at a more affordable cost. While these findings would be novel and important in improving clinical measurements of soft tissue, there are several major issues the authors need to address.

Experimental design

No comment.

Validity of the findings

No comment.

Additional comments

The authors should provide stronger justification of using the PACT compared to the MyotonPro. The data presented demonstrated that the MyotonPro was more reliable for two layer measurements compared to the PACT. While the authors state the PACT is sufficient, the authors should provide additional context (cost of device, hours of training required, etc.) to justify the use of a less reliable device.

In line 187, the authors state “suitability for the daily practice…” as a disadvantage for the MyotonPro. However, the authors demonstrated the MyotonPro has a superior ICC, indicating it is easier to use more reliably. The authors should remove this justification as it is not supported by the data.

Figure 2 should have correlation coefficient on the graph and in the figure description.

A similar figure as Figure 2 should be created for the two-layer testing results. Additionally, the authors should clarify how many measurements were taken for each different layer. As the Methods reads, it appears a single measurement (3 measures averaged to 1 measure) per examiner was taken for each stiffness layer. However, a correlation value is generated for each ‘stiffness’ and it’s unclear how that value was calculated from the single measurement.

Line 163-164 states the correlation between the PACT and MytonPRO was r=0.87. This should be graphically shown as well.

Figure 3 description should include a description of the results.

·

Basic reporting

Please consider reducing the use of non-common abbreviations in your manuscript, as it makes it harder to read and interpret your results. At times, you also mix up STM and SMT.

The introduction largely glosses over using ultrasound shear wave elastography to investigate myofascial tissue stiffness, including validity studies that compare it to the MyotronPRO.

Defining early if the stated goal is to use PACT to measure muscle stiffness or fascial tissue is needed - the use of 'myofascial tissue' here is vague.

No hypotheses are stated, which makes it hard to judge if the proposed statistical methods are appropriate.

Experimental design

Overall, I commend the authors for their study design, as it a strong comparative investigation of two technologies (one new to the market) using phantoms. Most of my concerns below are clarifying points, and not overall weakness in their approach.

Different stiffness levels are investigated, which strengthens their study design. However, there is no justification for why this particular range of values was selected. Is there prior evidence that can be referred to that justifies that your interested myofascial tissues have a Young Modulus between 24-166 kPa? (Also, using commas instead of periods in Table 1 is distracting).

For the two-layer phantom setup (Line 111), the authors need to verify the thickness of each layer. They previously discussed a 3-mm and 10-mm phantom, but its not clear if the 3-mm phantom was ever used in the experimental setup.

Why are you using the PACT sense summary score rather than the other advertised metrics from this device (e.g., stiffness, dampening) that would be more indicative of the viscoelastic properties?

Validity of the findings

The presentation of Table 3 is confusing - the methods suggest the 2nd layer of the phantom is the variable stiffness layer, but its presentation on Table 3 suggests it is the 1st layer that is variable. The interpretation of these results would be helpful if visuals of the data (similar to the single-layer phantom in Figure 2) could be provided.

It's unclear if Bland-Altman plots (Figure 3) were completed on the single layer or dual layer phantom.

The discussion section lacks an exploration of existing literature and where these results fit it. It's great that you did this work, but do the results make sense in the context of other validity studies using the MyotronPro or elastography?

What are the potential ramifications of the PACT device not working well with lower-stiffness tissues?

The conclusion statement is not well stated - its a real mouthful at the beginning "In sum, both tested STM"

---

## Round 0.2 · Minor Revisions

Please address Reviewer 2's comments

Reviewer 1 ·

Basic reporting

The authors have sufficiently addressed reviewer comments.

Experimental design

No comment

Validity of the findings

No comment

·

Basic reporting

I don’t feel the reviewers fully addressed my initial concern about the use of 'myofasicial tissue' being quite vague at the beginning of their manuscript. I think defining what they mean by myofascial tissue early (around line 47) is crucial for a naïve reader to understand their approach. They mean the stiffness of the myofascial unit (muscle + fascia), but that isn’t abundantly clear until the 4th paragraph.

Experimental design

No comment

Validity of the findings

It is unclear why Appendix Figure 1 is placed in an appendix and not presented in the main manuscript. Honestly, it is the most visually appealing figure/table in the study, and it makes it much easier to interpret their methods and resultant findings than the provided tables.

---

## Round 0.3 · accepted · Accept

The suggestions by the reviewers in the last review cycle have been addressed.